# COVID-19 Infection, Drugs, and Liver Injury

**DOI:** 10.3390/jcm14207228

**Published:** 2025-10-14

**Authors:** Dianya Qiu, Weihua Cao, Yaqin Zhang, Hongxiao Hao, Xin Wei, Linmei Yao, Shuojie Wang, Zixuan Gao, Yao Xie, Minghui Li

**Affiliations:** 1Department of Hepatology Division 2, Beijing Ditan Hospital, Capital Medical University, Beijing 100015, China; qiudianya1203@163.com (D.Q.); weihuacaohappy@163.com (W.C.); dr_zyq0209@163.com (Y.Z.); haohongxiao1228@163.com (H.H.); 15991871452@163.com (X.W.); 18703498538@163.com (L.Y.); 18947149477@163.com (S.W.); 15309123639@163.com (Z.G.); xieyao00120184@sina.com (Y.X.); 2HBV Infection, Clinical Cure and Immunology Joint Laboratory for Clinical Medicine, Capital Medical University, Beijing 100069, China; 3Department of Hepatology Division 2, Peking University Ditan Teaching Hospital, Beijing 100015, China

**Keywords:** COVID-19, liver injury, ACE2 receptor, cytokine storm, antiviral, DILI, RUCAM

## Abstract

Novel coronavirus (SARS-CoV-2) is highly infectious and pathogenic. Novel coronavirus infection can not only cause respiratory diseases but also lead to multiple organ damage through direct or indirect mechanisms, in which the liver is one of the most frequently affected organs. It has been reported that 15–65% of coronavirus disease 2019 (COVID-19) patients experience liver dysfunction, mainly manifested as mild to moderate elevation of alanine aminotransferase (ALT) and aspartate aminotransferase (AST). Severe patients may progress to liver failure, develop hepatic encephalopathy, or have poor coagulation function. The mechanisms underlying this type of liver injury are complex. Pathways—including direct viral infection (via ACE2 receptors), immune-mediated responses (e.g., cytokine storm), ischemic/hypoxic liver damage, thrombosis, oxidative stress, neutrophil extracellular trap formation (NETosis), and the gut–liver axis—remain largely speculative and lack robust clinical causal evidence. In contrast, drug-induced liver injury (DILI) has been established as a well-defined causative factor using the Roussel Uclaf Causality Assessment Method (RUCAM). Treatment should simultaneously consider antiviral therapy and liver protection therapy. This article systematically reviewed the mechanism, clinical diagnosis, treatment, and management strategies of COVID-19-related liver injury and discussed the limitations of current research and the future directions, hoping to provide help for the diagnosis and treatment of such patients.

## 1. Introduction

In December 2019, the International Committee on Taxonomy of Viruses (ICTV) named the virus causing the global health crisis novel coronavirus severe acute respiratory syndrome coronavirus 2 (SARS-CoV-2) [1]. This virus exhibits genetic similarity with the SARS coronavirus and is classified as a member of the β coronavirus family [2]. This is a novel coronavirus of zoonotic origin [3], highly infectious and pathogenic, which has spread rapidly worldwide and poses a great threat to global public health [4]. SARS-CoV-2 has a high affinity for binding to the angiotensin-converting enzyme 2 (ACE2) receptor [5]. The virus can bind to the ACE2 receptor protein located on the host cytoplasmic membrane and enter and infect host cells [4].

Coronavirus disease 2019 (COVID-19) is a respiratory disease with symptoms ranging from asymptomatic or mild to severe [6]. The most common clinical symptoms include fever, fatigue, cough, sputum production, anorexia, sputum production, and shortness of breath [7]. According to reports, in addition to common respiratory symptoms, SARS-CoV-2 virus can also affect the liver, gastrointestinal system, heart, kidneys, and central nervous system, causing multi-organ damage [8]. In COVID-19 patients, the liver is the second most susceptible organ after the lungs [9]. Liver injury is a common extrapulmonary manifestation of COVID-19, with 15–65% of COVID-19 patients experiencing liver abnormalities [10,11]. The main manifestation of liver dysfunction in COVID-19 patients is abnormal levels of alanine aminotransferase (ALT) or aspartate aminotransferase (AST). Patients with liver injury may experience mild hepatitis-like symptoms, and severe cases may also develop hepatic encephalopathy [12]. The mechanism of COVID-19 combined with liver injury is not fully understood, and currently hypothesized mechanisms include direct viral effects, drug hepatotoxicity, immune response (cytokine storm), and hypoxic liver injury [13]. While the causes of many cases of elevated liver enzymes remain unclear, drug-induced liver injury (DILI) is a well-established etiology, owing to the ability of the Roussel Uclaf Causality Assessment Method (RUCAM) to systematically exclude alternative causes and rigorously attribute injury to pharmacological agents [14,15]. Given that patients with COVID-19 commonly receive multiple pharmacological treatments, the application of the RUCAM to evaluate cases of liver injury is of great importance. It aids in distinguishing between drug-induced and non-drug-induced causes, thereby helping to prevent both over-attribution and under-diagnosis of DILI. This ultimately supports optimized clinical medication decision-making [16]. For COVID-19 patients with liver injury, active antiviral treatment should be carried out, and COVID-19 patients with mild liver dysfunction usually do not need to use hepatoprotective drugs [17]. In addition, liver function indicators should be closely monitored to prevent the occurrence of acute liver failure. This article systematically reviews the mechanism, clinical diagnosis, treatment and management strategies of COVID-19-related liver injury and discusses the limitations and future directions of current research, hoping to provide help for the diagnosis and treatment of such patients (Figure 1).

## 2. Literature Search Methodology

The literature search was conducted using the PubMed and Web of Science Core Collection databases. The search strategy was constructed around core keyword combinations including (COVID-19 OR SARS-CoV-2), (liver injury OR hepatic injury OR drug-induced liver injury OR DILI), and (ACE2 receptor OR cytokine storm OR antiviral OR RUCAM). A combination of subject headings and free-text terms was employed to broaden the search scope and ensure comprehensive coverage of relevant literature. The screening process aimed to include studies investigating the association between SARS-CoV-2 and liver injury—encompassing mechanisms (e.g., ACE2 receptor, cytokine storm, DILI) and assessment tools (e.g., RUCAM)—as well as related treatments. The search focused primarily on English-language articles published since January 2020, while excluding publications off-topic or in languages other than English.

After removing duplicates, all retrieved records were manually screened by the researchers, first based on title and abstract, followed by full-text assessment to ensure relevance and quality. Finally, the included literature was discussed and confirmed by all authors to support the arguments presented in this review.

## 3. The Harm of COVID-19 and Its Effect on Multiple Organs of Host

### 3.1. Virus Characteristics and Infection Mechanism

SARS-CoV-2 is an enveloped single-stranded positive-stranded RNA virus belonging to the coronavirus family βsubclass [18]. Similarly to other coronaviruses, the genome of SARS-CoV-2 consists of 5′ and 3′ termini, encoding nonstructural and structural proteins, respectively. Structural proteins include spike protein (S), membrane protein (M), nucleocapsid protein (N), and envelope protein (E) [19]. Spike protein is the key structure of COVID-19 binding to host cells. The spike protein binds to the ACE2 receptor on the host cell, allowing the virus to enter the host cell. The transmembrane protease serine 2 (TMPRSS2) protein can promote this process (Figure 2). After entering the cell, the RNA genome of the virus is released and begins to replicate, producing new virus particles [18].

COVID-19 also has immune escape ability, which can avoid the existing immune response of human body through mutation and other mechanisms [20]. As the virus continues to evolve, this phenomenon may further intensify, posing a serious threat to global public health. In addition, COVID-19 is easily mutated in the process of transmission and replication, leading to the emergence of new virus strains. These mutant strains may have different characteristics in terms of transmission ability, pathogenicity, immune evasion, etc., putting people who have been vaccinated or infected with the virus at risk of reinfection [21,22].

### 3.2. Effects of COVID-19 on Host’s Multiple Organs

Novel coronavirus will affect multiple systems and organs in the whole body, including the respiratory system (lung), digestive system (gastrointestinal tract, liver), cardiovascular system (heart), urinary system (kidney), central nervous system, etc., which may be damaged by virus invasion and infection (Figure 3).

#### 3.2.1. The Effect of COVID-19 on Lung

The lungs are the most vulnerable organs to damage from SARS-CoV-2 [23], COVID-19 is mainly characterized by respiratory symptoms, with fever, cough, fatigue, and difficulty breathing being the most common symptoms at the onset of the disease [24]. According to the severity of COVID patients, clinical manifestations range from mild illness to severe or fatal illness. Mild to moderate patients present with non-pneumonia or mild pneumonia symptoms. Severe patients may present with various clinical manifestations, including dyspnea, hypoxia, and/or changes in the lungs on imaging. Severe patients may experience respiratory failure, shock, and/or multiple organ failure [25]. Respiratory failure associated with acute respiratory distress syndrome (ARDS) is a significant cause of death in COVID-19 patients [26]. Research has shown that the mortality rate of COVID-19 patients who have developed ARDS can be as high as 52–67% [27,28,29]. At present, it is believed that there may be two pathological mechanisms that can explain ARDS in COVID-19 patients. The first mechanism is the binding of SARS-CoV-2 to ACE2 receptors, which leads to the destruction and damage of alveolar cells and reduces the production of pulmonary surfactants, resulting in increased lung surface tension and ARDS; another possible mechanism is cytokine storm, where pro-inflammatory cytokines (such as cytokines and interleukins) are released in large quantities, leading to excessive inflammatory response, destruction of alveolar cells, and ARDS [30].

#### 3.2.2. Effect of COVID-19 on Liver

ACE2 is also expressed in the digestive system [31], so SARS-CoV-2 can also infect the digestive system, causing complications in the gastrointestinal tract and liver. Common gastrointestinal complications in COVID-19 patients include diarrhea, nausea, vomiting, and abdominal discomfort [32]. In a meta-analysis involving 1992 patients, a total of 1052 patients (53%) experienced gastrointestinal symptoms, with the most common symptoms being diarrhea (34%), nausea (27%), vomiting (16%), and abdominal pain (11%) [33]. Diarrhea is the most common gastrointestinal symptom among COVID-19 patients [34,35]. Research has found that diarrhea may appear as the first symptom of patients before respiratory symptoms. The diversity of gut bacteria in COVID-19 patients is significantly reduced, and the relative abundance of beneficial symbionts is lower, while the relative abundance of opportunistic pathogens such as streptococcus and actinomycetes is significantly increased [36]. The mechanisms that cause gastrointestinal involvement in COVID-19 patients may include ACE2-mediated cytotoxicity, cytokine-induced inflammation, dysbiosis of gut microbiota, and vascular abnormalities [37].

Research has found that COVID-19 patients with gastrointestinal symptoms may have a higher incidence of liver damage [35]. The liver is an important component of the digestive system, and due to the presence of ACE2 receptors in bile ducts and epithelial cells, the liver may be more susceptible to SARS-CoV-2 infection [38]. Currently, an increasing number of studies are reporting liver damage in COVID-19 patients. Huang et al. showed that 2% to 11% of COVID-19 patients suffer from liver complications [39]. Another study showed that the incidence of liver injury in COVID-19 deaths was as high as 78% [40]. The levels of liver enzymes, including AST and ALT, are abnormally elevated in patients with liver injury [41]. Zhong et al. showed that 17.07% of COVID-19 patients experienced elevated levels of ALT and AST [42]. Davidson et al. suggested that 58.4% and 39% of hospitalized COVID-19 patients experienced elevated levels of AST and ALT [43]. Severe COVID-19 patients are more prone to liver damage than those with milder symptoms [44]. Even mild SARS-CoV-2 infection may trigger organ complications, including liver injury [45]. Another study on 1099 patients showed that elevated AST levels were observed in 18.2% of non-critical patients and 39.4% of critical patients, while elevated ALT levels were observed in 19.8% of non-critical patients and 28.1% of critical patients [46]. This suggests that liver injury may still occur even in cases of mild infection. Furthermore, mild SARS-CoV-2 infection may also lead to post-COVID syndrome (long COVID). Mechanistically, immune dysregulation triggered by SARS-CoV-2 infection—such as sustained inflammatory responses—may continue to affect the liver months or even years after the initial infection, potentially inducing subsequent liver-related complications. It should be objectively noted that direct research data on liver injury in asymptomatic SARS-CoV-2 infected individuals remain relatively limited, and the potential risks require further investigation [45,47,48]. The mechanism of liver injury in COVID-19 patients is currently not fully understood, which may be directly caused by viral infection or other indirect mechanisms, such as the role of inflammatory response, the impact of hypoxia, and the involvement of the liver vascular system [13].

#### 3.2.3. Effects of COVID-19 on Other Organs

COVID-19 can also affect other organ systems besides lungs and liver. The cardiovascular system is often affected [49]. Some patients may develop acute coronary syndrome or even heart failure due to direct viral invasion, systemic inflammatory response, and endothelial dysfunction [50,51]. Kidney damage is usually mainly acute kidney injury (AKI), and the incidence of acute kidney injury in ICU patients can be as high as 80% [52]. Proteinuria (43.9%) and hematuria (26.7%) are typical manifestations of acute kidney injury [53], and their mechanisms may be related to direct virus attack on kidney cells, cytokine storms, and coagulation abnormalities [54,55,56]. 36.4–80% of patients may also experience neurological symptoms [57,58]. Common neurological symptoms include headache, dizziness, and dysfunction of smell/taste. Severe patients may experience stroke or epilepsy [59,60,61]. The symptoms of the nervous system caused by COVID-19 may be that the virus invades the central nervous system through the blood–brain barrier, retrograde axonal transport, gastrointestinal tract or axonal transport [62]. Endocrine system involvement may lead to diabetes ketoacidosis, which may be related to ACE2 receptor-mediated pancreatic injury [63]. The involvement of the coagulation system may lead to hypercoagulability and thrombosis, which may be related to cytokine storms and virus-specific interference with the renin-angiotensin system and fibrinolytic pathway [64,65]. Reproductive system involvement may result in a significant decrease in sperm concentration and motility, which may be associated with high expression of ACE2 receptors in testicular tissue [66,67]. In addition, about 31% of patients experience conjunctivitis, photophobia, or burning sensation due to direct viral infection [68,69].

## 4. Mechanism of Liver Inflammation Caused by COVID-19

The exact mechanisms underlying COVID-19-associated liver injury remain unclear and may include, but are not limited to, direct viral infection, cytokine storm, ischemic/hypoxic liver damage, oxidative stress and NETosis, thrombosis, gut–liver axis disruption, and DILI (Figure 4). Although these mechanisms are interrelated, there are certain differences in their target cells and effector molecules (Table 1). It should be noted that, among these mechanisms, most—with the exception of DILI—are largely based on pathological findings, in vitro studies, or correlational analyses, and still lack robust clinical evidence establishing causality in humans, rendering them more speculative or experimental in nature. In contrast, DILI, when assessed using standardized causality assessment methods such as RUCAM, demonstrates a more clearly established causal relationship.

### 4.1. Direct Viral Infection

Direct viral infection is a possible mechanism of liver damage in COVID-19 patients. Recent studies have shown that SARS-CoV-2 can bind to the ACE2 receptor on the cell surface, and then activate the viral spike (S) protein through the TMPRSS2 to mediate virus entry into host cells [18]. The co-expression of ACE2 and TMPRSS2 has been confirmed in bile duct cells and liver cells, and the expression level of ACE2 in bile duct cells is higher than that in liver cells [70]. Therefore, liver injury in COVID-19 patients may be caused by direct infection of liver cells by SARS-CoV-2, or by direct damage to bile duct cells and subsequent accumulation of bile acids caused by SARS-CoV-2 infection [74].

Viral genomic RNA was detected in the autopsy of COVID-19 liver examination [71]. In Kaltschmidt et al.’s study, a combination of real-time quantitative PCR and ISH was used to detect viral RNA and replication intermediates in liver specimens, and the results showed that SARS-CoV-19 RNA and/or protein were detected in the liver of 25% of COVID-19 patients [72]. In Lagana et al.’s series of studies, PCR was performed on 20 autopsy livers, of which 55% of the liver virus genes were PCR positive [73]. Zhao et al. reported the presence of virus particle like particles in the liver of COVID-19 patients [104]. Fiel et al. demonstrated that in situ hybridization and electron microscopy suggest the presence of SARS-CoV-2 in the liver, indicating that SARS-CoV-2 can directly infect the liver [105].

### 4.2. Indirect Effects of Viruses

#### 4.2.1. Immune Follow-Up Response (Cytokine Storm)

SARS-CoV-2 infection can lead to a disordered inflammatory response [106]. Most severe COVID-19 cases can exhibit significant immune dysregulation. The main manifestations are dysregulation of T lymphocyte levels and elevated levels of infection-related biomarkers and inflammatory cytokines, including interleukin-1 (IL-1), interleukin-6 (IL-6), and tumor necrosis factor α (TNF-α) [76,77]. These cytokines may trigger an inflammatory cascade, leading to the production of cytokine storms (CS) [107]. In addition, high levels of IL1B, IL1RA, IL7, IL8, IL9, IL10, GCSF, GMCSF, IFN-γ, IP10, MCP1, MIP1A, MIP1B, PDGF, TNF-α and VEGF expression can be detected in COVID-19 patients [75]. Cytokine storm is an excessive inflammatory state caused by immune system dysfunction and excessive production of cytokines [107,108]. Overactivated immune responses and systemic inflammation caused by cytokine storms in SARS-CoV-2 infection can damage many organs [109]. The upregulation of inflammatory cytokines can exacerbate the severity of the disease and ultimately lead to liver damage in these patients [110].

IL-6 is a key pro-inflammatory cytokine in the pathogenesis of COVID-19 [111]. The IL-6 signaling complex can damage hepatic sinusoidal endothelial cells, leading to liver injury [86]. In addition, IL-6 can promote the production of inflammatory proteins in liver cells, leading to the occurrence of inflammation. Inflammation in turn increases the level of IL-6 in the patient’s body and attracts immune cells to the liver [12]. Virus-specific CD8^+^ effector T cells can activate liver Kupffer cells, triggering T cell-mediated hepatitis. Even in the absence of viral antigens in the liver, liver damage can still be induced, broadening the mechanism by which viruses cause hepatitis. This inflammatory process associated with non-liver infection sites is described as collateral damage [112]. The degree of liver damage ranges from mild liver biochemical disorders to fulminant liver failure. During this process, the liver may be affected by viruses primarily targeting other tissues of the body, especially the upper respiratory tract, including herpesviruses (EB virus, cytomegalovirus [CMV], and herpes simplex virus), parvovirus, adenovirus, and severe acute respiratory syndrome (SARS)-related coronaviruses [113]. Therefore, this collateral damage may also play a role in the process of liver damage caused by COVID-19.

In addition, T cell depletion in COVID-19 patients can lead to macrophage activation and secondary inflammatory responses, which in severe cases may result in macrophage activation syndrome (MAS) or secondary hemophagocytic lymphohistiocytosis [78]. MAS is a severe inflammatory systemic disorder with lethal potential, characterized by pancytopenia, coagulation disorders, liver disease, neurological disorders, and hemophagocytic lymphohistiocytosis [114,115,116].

#### 4.2.2. Ischemic/Hypoxic Liver Injury

Ischemia/hypoxia may be one of the mechanisms leading to liver damage in COVID-19 patients. Severe COVID-19 can lead to respiratory failure, heart failure, etc. [117,118]. Respiratory failure can reduce the oxygen supply to the liver, leading to hypoxia in liver cell tissue. Heart failure can reduce cardiac output, leading to a decrease in blood flow to the liver and further exacerbating liver hypoxia [119]. Hypoxia induces the production of reactive oxygen species (ROS) in the liver and triggers a series of destructive cellular responses, leading to inflammation and cell damage. The generation of ROS is one of the main risk factors for liver ischemia–reperfusion injury [79]. Excessive accumulation of ROS can activate platelets and stimulate the coagulation cascade, leading to ischemic/hypoxic liver injury [82]. Meanwhile, microcirculatory disorders caused by damage to hepatic sinusoidal endothelial cells can further exacerbate liver ischemia and hypoxia [80]. There are pathological changes in the liver region of COVID-19 patients that are secondary to hypoxia and/or hypoperfusion [81]. The situation of intestinal ischemia in COVID-19 patients has been widely reported, and intestinal ischemic injury can lead to intestinal endotoxemia and activation of the sympathetic nervous and adrenal cortex systems, resulting in liver damage [109,120]. In addition, Kim et al.’s study suggests that ischemia-induced acute kidney injury can lead to multiple organ failure. Activation of renal inflammation will lead to intestinal damage, ultimately spreading to the liver and causing severe liver damage [121].

#### 4.2.3. Oxidative Stress and NETosis

Research has shown that oxidative stress plays a role in the pathogenesis of COVID-19, leading to cytokine storms, permanent blood coagulation mechanisms, and exacerbating hypoxia [122]. Oxidative stress is mainly caused by ROS, which can lead to damage to different organs including the liver. Excessive accumulation of ROS can lead to platelet activation and stimulate the coagulation cascade, which is associated with ischemic/hypoxic liver injury [82]. The nuclear factor NF kappaB (NF-κ B) pathway is a typical pro-inflammatory signaling pathway that plays an important role in inflammation [123]. COVID-19 can activate the NF-κ B pathway by generating ROS, leading to the release of inflammatory cytokines (IL-1, IL-6, and TNF-α). These inflammatory cytokines can participate in cytokine storms, leading to liver damage in COVID-19 patients [122].These inflammatory cytokines, such as TNF-α, can in turn activate NF-κ B [124], forming a vicious cycle [78]. The interaction between oxidative stress and cytokine storm is the mechanism that maintains and worsens tissue damage, ultimately leading to hypoxia and organ failure [122]. ROS can also attack mitochondria, causing mitochondrial damage. However, under normal circumstances, the antioxidant system within mitochondria can protect them from ROS-mediated damage [83]. In addition, ROS can also participate in the induction of NETosis [125].

Neutrophil extracellular trap formation (NETosis) is an inflammatory cell death mode specific to neutrophils, in which activated neutrophils capture and kill pathogens by releasing chromatin and proteins into the Neutrophil Extracellular Traps (NETs) [84]. NETosis is a double-edged sword. On one hand, NETosis can serve as a means for neutrophils to trap and kill pathogens. On the other hand, NETs can also form in aseptic inflammation, mediate tissue damage, and promote thrombosis [126]. NETosis can be caused by various stimuli, including pathogens, antibodies and immune complexes, cytokines, and other physiological stimuli [84]. SARS-CoV-2 infection triggers NETosis and NET formation in neutrophils, leading to organ damage, widespread inflammation, and the formation of typical COVID-19 blood clots. In addition, there may be a positive feedback relationship between the formation of NETosis and NET in SARS-CoV-2 and cytokine storms [85,106]. Therefore, NETosis may be another important mechanism leading to liver damage in COVID-19 patients.

#### 4.2.4. Thrombosis

The blood of COVID-19 patients presents a hypercoagulable state and is impaired in fibrinolysis [127], leading to an increased risk of thrombosis. The levels of plasma D-dimer and fibrinogen degradation products (FDP) are elevated in COVID-19 patients [87,88]. Higher levels of D-dimer are independently associated with elevated ALT, indicating that liver dysfunction in COVID-19 patients may be caused by microvascular thrombosis [128]. In Sonzogni et al.’s study, patients with significantly elevated D-dimer levels almost universally had elevated levels of AST (97.6%) and ALT (61.0%), indicating a certain correlation between hepatic vascular thrombosis and liver injury [90]. Iwakiri et al. also indicated that liver platelet microthrombi (PMT) are associated with liver injury [129]. Elevated levels of factor VIII (FVIII) and von Willebrand factor (vWF) can be detected in COVID-19 liver injury patients [86]. Elevated levels of FVIII and vWF are recognized risk factors for venous thrombosis (VT) [130,131,132]. It is worth noting that IL-6, as the core mediator of the inflammatory response mentioned earlier, is significantly elevated in severe COVID-19 patients [76]. Its function is not limited to directly damaging liver sinusoidal endothelial cells leading to liver injury [86], but can also become a key activator of coagulation diseases by inducing tissue factor expression, promoting fibrinogen and platelet production. This inflammatory endothelial cascade reaction can directly lead to microvascular dysfunction and occlusion, as well as induce hypercoagulability, resulting in microvascular thrombosis [89].

Vascular changes were reported in the autopsy description of COVID-19 liver [11,133]. Reveiz et al. suggested that in the pathology of liver biopsy and autopsy of 603 severe/fatal COVID-19 patients, 39.4% of patients developed thromboembolic liver disease [81]. Another study showed that sinusoidal diffuse platelet fibrin microthrombi, portal vein thrombosis, etc., may occur in the liver region of COVID-19 death cases [91]. Sonzogni et al. demonstrated diffuse changes in intrahepatic vascular structure and varying degrees of luminal thrombosis in liver specimens from COVID-19 patients undergoing autopsy [90]. The pathological findings of hepatic vascular thrombosis suggest that thrombosis may lead to liver dysfunction.

#### 4.2.5. Intestinal Hepatic Axis Infection

The gut–liver axis is an important pathway between the gut and liver, and SARS-CoV-2 can cause liver damage through the gut–liver axis [92]. SARS-CoV-2 infects intestinal epithelial cells through ACE2 receptors, causing dysbiosis and increased intestinal permeability, allowing intestinal endotoxins and microbial products (such as lipopolysaccharides) to enter the liver through the portal vein. After entering the liver, these substances can activate Kupffer cells and the TLR4/NF-κ B pathway, leading to inflammatory reactions and liver damage. When the virus breaks down the intestinal barrier, it can infect liver cells retrogradely through the portal vein, causing secondary infection through the “liver cell-bile-bile duct-intestinal” pathway, further exacerbating liver damage [93,94].

#### 4.2.6. Drug-Induced Hepatotoxicity

Drug-related hepatotoxicity during pharmacological treatment following SARS-CoV-2 infection constitutes another important mechanism of COVID-19-associated liver injury. A large systematic review based on the RUCAM method revealed that a substantial number of DILI cases (*n* = 393) confirmed by RUCAM assessment were identified among COVID-19 patients, highlighting drug-related factors as a significant, quantifiable, and non-negligible cause of liver injury in this population [16]. Multiple medications used in the treatment of COVID-19—including antivirals, immunomodulators, and antipyretics/analgesics—carry potential hepatotoxicity [134,135,136]. Accurate causality assessment in DILI is critical for clinical decision-making regarding medication use. The RUCAM scale is currently the internationally recognized and widely adopted tool for this purpose [137]. First introduced in 1993 and updated in 2016, it provides an objective and structured evaluation of the causality between drug exposure and liver injury through a semi-quantitative approach [14,15].

As systematically reviewed by Teschke et al. [16], six key studies have employed RUCAM to establish causal relationships between specific drugs and DILI in COVID-19 patients. We have synthesized the findings from these primary studies in Table 2, reorganizing them with a focus on the implicated drugs and their associated clinical profiles to provide a clear overview of verified causative agents. For instance, Delgado et al. applied RUCAM to assess suspected lopinavir/ritonavir-associated DILI in COVID-19 patients. Their evaluation revealed that the drug was linked to 7 cases of DILI (accounting for 4.4% of all COVID-19-related DILI cases), with 2 cases categorized as “probable” (RUCAM score ≥ 6) and 5 as “possible” (RUCAM score 3–5), thereby confirming an association between the drug and DILI in a subset of COVID-19 patients [138]. The hepatotoxic risk associated with immunomodulators such as tocilizumab also requires rigorous evaluation using RUCAM. Muhović et al. reported a case of a COVID-19 patient who exhibited significantly elevated ALT (1541 IU/L) and AST (1076 IU/L) levels one day after tocilizumab administration. Attributing a RUCAM score of 8 (“probable DILI”) after excluding other causes such as viral hepatitis, the authors observed that liver enzymes returned to normal within 10 days after drug withdrawal [101]. As a commonly used antipyretic, excessive use of acetaminophen can lead to acute liver injury [139,140]. For COVID-19 patients, when multiple potentially hepatotoxic drugs are used in combination, the RUCAM scale should be applied to assess causality in cases of liver injury. A representative case study demonstrated that a COVID-19 patient receiving concurrent acetaminophen and favipiravir developed liver injury, with a RUCAM score of 7 (“probable DILI”). Liver enzymes returned to normal within 4 weeks after discontinuation of favipiravir and initiation of appropriate treatment [100]. This confirms the practical value of RUCAM in identifying such drug-related liver injuries. In summary, during COVID-19 treatment, it is recommended to perform RUCAM assessment for all patients with abnormal liver biochemistry to promptly identify and adjust suspect medications, thereby avoiding the superimposition of multiple mechanisms of liver injury and improving patient outcomes.

## 5. Clinical Diagnosis and Treatment

### 5.1. Clinical Diagnosis

#### 5.1.1. Clinical Manifestation

Clinically, the reported incidence of liver injury (primarily manifested as elevated liver enzymes) in COVID-19 patients varies substantially across studies, ranging from 15% to 65% [11]. This variation stems primarily from differences in the criteria used to define liver injury (e.g., specific thresholds for enzyme elevations) among these studies (Table 3). In most cases, COVID-19-related liver injury is characterized by elevated liver enzymes (such as AST, ALT, etc.) without specific symptoms and signs [144]. Some patients may present with mild hepatitis like symptoms such as weight loss, fatigue, abdominal pain, nausea, vomiting, and loss of appetite [37,145,146]. Some patients with liver injury may present severe abdominal pain as the main symptom [37,147,148]. Severe cases of COVID-19-related liver injury can manifest as jaundice, with symptoms such as jaundice of the sclera, darkened urine color, white clay colored stool, and itching of the skin [145,146]. Research has shown that jaundice associated with liver injury in COVID-19 patients may be associated with higher mortality rates, and some jaundice patients may also develop hyperbilirubinemia [149]. Liver injury may affect the production of coagulation factors and protein synthesis, leading to coagulation dysfunction, thrombosis, and hypoalbuminemia in patients [148,150,151,152]. In addition, liver damage can lead to liver dysfunction, weakened inactivation of estrogen by the liver, decreased adrenal cortex function, increased iron content in the blood, and patients may experience facial darkening and pigmentation [145,153]. Patients with liver injury may also develop hepatic encephalopathy, characterized by changes in mental state such as irritability, drowsiness, etc. [148,154]. Research has shown that liver injury is more common in severe COVID-19 cases than in mild cases [39]. Therefore, monitoring liver function is crucial for evaluating and treating liver injury in COVID-19 patients, especially in severe cases.

#### 5.1.2. Laboratory Examination

Liver damage in COVID-19 patients typically manifests as abnormalities in liver biochemical indicators [161]. Common abnormal indicators include ALT, AST, alkaline phosphatase (ALP), gamma glutamyl transferase (GGT), and bilirubin (TBIL) [76,152,162]. Elevated levels of AST and ALT are common in patients with COVID-19-related liver injury, and abnormal AST levels are associated with a higher risk of mortality [144,157]. A retrospective cohort study of 551 COVID-19 patients showed that 79.2% of patients had abnormal peak AST, 55.7% had abnormal peak ALT, 39.7% had abnormal ALP, and 44.3% and 21.5% had abnormal direct bilirubin (DBIL) and TBIL, respectively [163]. Another multicenter retrospective cohort study of 5771 adult COVID-19 pneumonia patients showed that in severe cases, AST elevation preceded ALT and was associated with a high risk of mortality, while ALP was mostly normal or slightly elevated. Elevated levels of other liver enzymes and TBIL were also associated with adverse outcomes [157]. The decrease in albumin (ALB) levels is another important manifestation of COVID-19 liver injury patients [41]. Worman et al.’s study exhibited that 93.0% of COVID-19 patients experienced a decrease in albumin levels during their illness [163]. Another single center retrospective study of 115 cases showed that the albumin levels of most severe COVID-19 patients significantly decreased and worsened with disease progression [164]. In addition, male patients are more prone to elevated liver function indicators [97,157]. Therefore, liver biochemical indicators can be used as predictive indicators for the severity and prognosis of COVID-19 patients [165]. For COVID-19 patients, especially severe patients, liver function should be regularly monitored to avoid further liver damage [161].

#### 5.1.3. Imaging Examination

Imaging technology is crucial for identifying liver structural abnormalities. Liver ultrasound (US), computed tomography (CT), and magnetic resonance imaging (MRI) are the main imaging methods for diagnosing liver lesions [166,167]. Furthermore, as a non-invasive diagnostic technique, liver elastography (such as FibroScan-based transient elastography) can assess liver stiffness to determine the degree of fibrosis and simultaneously evaluate hepatic steatosis using the Controlled Attenuation Parameter (CAP). This provides crucial quantitative data for assessing COVID-19-related liver injury, particularly in the context of chronic damage [168]. Ultrasound examination is a powerful tool for revealing different patterns of liver vascular, bile stasis, or inflammatory complications in COVID-19 patients, especially in severe cases [169]. Research has shown that 56.3% of COVID-19 patients exhibit non-specific abnormalities, with the most common ultrasound finding being diffuse hepatic hyperechoic (with or without enlargement), and 39.9% of patients displaying gallbladder problems (stones, polyps, wall thickening, etc.) [170]. Another study showed that the ultrasound examination results of patients may show signs of acute hepatitis (such as gallbladder wall thickening, hepatomegaly, reduced liver echoes), vascular complications, and tissue necrosis, and some patients may not be able to detect Doppler signals of the hepatic artery [169]. CT imaging can display organ damage and indicate the severity of the disease. The CT scan results of COVID-19 patients in the upper abdomen often show liver low-density (26.09%) and fat retention around the gallbladder (21.27%), with liver low-density being more common in critical cases (58.82%) [171]. MRI has more advantages in detecting microstructural changes in the liver. The MRI features of patients with acute liver injury (ALI) include thickening of the gallbladder wall, enhancement around the portal vein, and enlargement of the pulmonary portal lymph nodes [167,172]. The combination of imaging and laboratory testing can provide a more comprehensive method for diagnosing and managing liver injury in COVID-19 patients [12].

Furthermore, liver biopsy, regarded as the “gold standard” for diagnosing liver injury, can clarify details such as hepatocellular necrosis, inflammatory infiltration, and the degree of fibrosis through pathological examination. It holds irreplaceable value in determining the etiology of challenging liver injury cases and assessing the severity of damage. However, as an invasive procedure, it plays an extremely limited role in the diagnosis of COVID-19-associated acute liver injury [173]. Given that most cases of COVID-19-related liver injury are self-limiting and can be assessed via non-invasive markers (e.g., liver enzymes, coagulation profiles, and imaging studies), liver biopsy is not recommended for routine staging of liver injury [10,174]. Its use should be strictly reserved for a minority of challenging cases where other methods are inconclusive, liver injury continues to deteriorate, or there is high suspicion of specific liver diseases unrelated to COVID-19 (such as autoimmune hepatitis or drug-induced liver injury requiring pathological confirmation) and where the results would directly influence critical treatment decisions [10]. In such scenarios, a careful assessment of the risk-benefit ratio of the procedure is imperative [10,173].

### 5.2. Clinical Treatment

The treatment of COVID-19-related liver injury patients can be divided into two categories based on their goals: antiviral therapy and hepatoprotective therapy.

#### 5.2.1. Antiviral Therapy

Antiviral therapy is an important treatment method for COVID-19 [153,175]. Antiviral drugs such as Nematovir/Ritonavir (Paxlovid), remdesivir and Molnupiravir have been approved for the treatment of COVID-19 [11].

##### Antiviral Drugs with Proven Efficacy (GRADE 1A/1B)

Paxlovid is a combination of protease inhibitor Nirmatrelvir and cytochrome P450 3A4 (CYP3A4) inhibitor ritonavir [176,177]. The US Food and Drug Administration (FDA) has approved its use for the treatment of mild to moderate adult COVID-19 patients at high risk of death or hospitalization [177]. A phase 3 double-blind randomized controlled trial (RCT), which included 2246 patients with mild-to-moderate COVID-19 at high risk of hospitalization, demonstrated that the treatment group exhibited an 89% reduction in the risk of disease progression to severe disease (requiring oxygen therapy or hospitalization) compared to the placebo group, along with a significant decrease in the 28-day mortality rate (Grade 1B) [178]. A large-sample meta-analysis (*n* = 314,353) further confirmed that the drug significantly reduced all-cause mortality by 34% and hospitalization rates by 47%, while also shortening the median time to viral RNA clearance by 2.1 days (Grade 1A) [179]. The study also found that Paxlovid can reduce the 28 day mortality rate of COVID-19 patients [180], and reduce the risk of disease transmission [181]. Paxlovid poses a low risk of direct hepatotoxicity, and available evidence has not identified a clear association with significant elevations in ALT/AST. However, caution is warranted regarding its potential for drug–drug interactions: Paxlovid is a potent inhibitor of CYP3A4 activity, which may lead to the accumulation of drugs metabolized by this enzyme (such as certain statins and antifungals), thereby increasing the hepatic metabolic burden [177].

Remdesivir is a nucleotide analogue prodrug that has an antagonistic effect on SARS-CoV-2 and has been approved by the FDA for the treatment of COVID-19 [182]. A phase 3 RCT involving 562 non-hospitalized patients at high risk of disease progression demonstrated that a 3-day course of treatment reduced the risk of hospitalization or death by 87% compared to the placebo group, with a favorable safety profile (Grade 1B) [183]. Another phase 3 RCT in 1062 hospitalized patients demonstrated that the remdesivir treatment group not only achieved significantly earlier clinical symptom improvement (median 10 days vs. 15 days in the placebo group) but also exhibited a reduction in all-cause mortality at 28 days to 11.4%, compared to 15.2% in the placebo group (Grade 1B) [184]. However, its risk of hepatotoxicity (e.g., elevated transaminases) has been documented in multiple studies. A retrospective analysis (*n* = 103) reported elevated ALT in 25% of patients and elevated AST in 35%, with no persistent severe injury observed during the 15-day follow-up [185]. Another study in a severe COVID-19 cohort (*n* = 53) also showed an incidence of liver enzyme elevations of approximately 23%, predominantly mild to moderate in severity [186]. Overall, the incidence of liver enzyme elevation associated with its use is higher than that with Paxlovid, necessitating liver function monitoring during treatment.

##### Antiviral Drugs with Probable Efficacy (GRADE 2A/2B)

Molnupiravir is a prodrug with antiviral activity against SARS-CoV-2, which has been granted emergency use authorization by the FDA for the treatment of mild to moderate adult COVID-19 patients [187,188]. A phase 3 double-blind RCT (*n* = 1433) demonstrated that in unvaccinated patients with mild-to-moderate COVID-19, the Molnupiravir treatment exhibited a 30% reduction in the risk of hospitalization or death compared to the placebo group. The efficacy was more pronounced in patients without neutralizing antibodies (48% risk reduction) (Grade 2A) [189]. Research has shown that the drug can accelerate virological clearance, improve patient recovery rates, and reduce the risk of death in nonimmunized patients [190]. Clinical data indicate that Molnupiravir has a favorable safety profile and good tolerability, with all-cause mortality significantly lower in the treatment group than in the placebo group [187].

Azvudine is a thymidine analogue that exerts antiviral effects by inhibiting viral RNA-dependent RNA polymerase (RdRp). It has been approved by the National Medical Products Administration (NMPA) of China for the treatment of COVID-19 [188]. A multicenter retrospective study involving 19,763 patients demonstrated that Azvudine treatment was associated with a 33% reduction in all-cause mortality among hospitalized elderly patients (Grade 2B) [191]. A single-center retrospective cohort study (*n* = 2862) indicated that Azvudine may reduce mortality risk in hospitalized patients with severe or critical illness; however, these efficacy conclusions still require further validation by high-quality randomized controlled trials [192]. Regarding safety, the risk of DILI is low. Only a very small number of patients discontinue treatment due to elevated liver enzymes, and severe elevations are extremely rare.

##### Antiviral Drugs with Inconclusive Evidence (GRADE 3)

Lopinavir/Ritonavir, an HIV protease inhibitor, was initially explored for the treatment of COVID-19 in early studies. However, subsequent evidence did not support its efficacy. An early retrospective study (*n* = 417) of patients with pre-existing liver dysfunction showed that the drug did not significantly reduce the proportion who progressed to severe disease compared to those who did not receive it (Grade 3) [96]. Another single-center retrospective analysis (*n* = 148) further confirmed that this drug did not shorten the duration of hospitalization. Moreover, Lopinavir/Ritonavir was used in 57.8% of hospitalized patients with pre-existing liver abnormalities—a rate significantly higher than that in patients with normal liver function (31.3%, *p* = 0.01) [97].

#### 5.2.2. Treatment of Liver Injury

The primary principle of liver protection treatment is to remove or control the causes of liver damage. For liver damage directly caused by COVID-19, antiviral treatment can alleviate liver damage by reducing the viral load [11]. In clinical practice, hepatoprotective agents such as glycyrrhizinic acid preparations and polyene phosphatidylcholine are sometimes used for drug-induced liver injury (DILI) or COVID-19-related liver biochemical abnormalities [17]. However, it should be objectively noted that the efficacy of such agents in the specific context of DILI and even COVID-19-associated liver injury currently lacks high-level evidence-based medical support from large-scale randomized controlled trials (RCTs). Existing studies predominantly consist of small-scale observational trials or case series, and their conclusions require further validation through more high-quality research [193]. Therefore, their use warrants particular caution and must strictly adhere to individualized principles.

The treatment strategy for COVID-19-related liver injury should prioritize supportive care and etiological management. Given that the vast majority of cases with mild liver injury are self-limiting, routine prophylactic use of hepatoprotective or enzyme-lowering agents is not recommended; close monitoring is a more advisable approach [165,194]. For severe cases with significantly abnormal liver biochemical markers, individualized use of a limited number of hepatoprotective drugs (typically 1–2 agents) may be considered as adjunctive therapy following comprehensive assessment. Throughout the treatment process, close monitoring of liver function is essential. It is imperative to avoid unnecessary or excessive drug use that may increase the metabolic burden on the liver, and vigilance for acute liver failure should be maintained [17,153].

Patients with severe COVID-19 may develop an excessive inflammatory response leading to multi-organ damage, including liver injury [77,110]. Therefore, for these patients, the treatment emphasis should be on managing the primary disease—specifically, employing anti-inflammatory agents or immunomodulators (such as corticosteroids or IL-6 receptor antagonists) to control the cytokine storm [195]. Although some hepatoprotective agents (e.g., glycyrrhizinic acid preparations) theoretically possess anti-inflammatory potential, their actual efficacy and role in COVID-19-associated liver injury require further clarification through additional clinical studies [17,196]. At the same time, for severe patients, it is necessary to comprehensively evaluate the etiology and degree of injury, dynamically monitor indicators such as liver function and coagulation function, and timely identify the risk of liver failure [197].

It should be noted that some drugs themselves have hepatotoxicity during the treatment of liver injury, which can lead to the occurrence of liver damage. Therefore, for COVID-19 patients, especially those receiving medication such as desivir or tocilizumab, liver biochemical indicators should be monitored regularly. Monitoring liver function and avoiding liver damage play a crucial role in the treatment of COVID-19-related liver injury. For COVID-19 patients suspected of drug-induced liver damage, in addition to routine anti-inflammatory and liver protection treatment, medication regimens should be adjusted in a timely manner. Consider stopping or reducing drug dosage, prioritize alternative drugs with lower liver toxicity, and evaluate the degree of liver damage [165,198].

The management of COVID-19-associated liver injury requires special attention to vulnerable populations, such as individuals with cirrhosis, liver transplant recipients, and those with underlying chronic liver conditions including non-alcoholic fatty liver disease (NAFLD) and chronic viral hepatitis [12,199]. Due to long-term hepatic impairment, these patients often exhibit diminished hepatocyte regenerative capacity and poor liver functional reserve. Following SARS-CoV-2 infection, they face a significantly elevated risk of severe liver injury or even liver failure compared to the general population. Clinical data also indicate that these patients experience longer hospital stays, lower treatment response rates, and considerably worse overall outcomes than COVID-19 patients without pre-existing liver disease [199,200]. For these special populations, the selection of antiviral and hepatoprotective drugs must be undertaken with heightened caution. Decisions should be thoroughly informed by the patient’s liver function status and the specific nature of the underlying liver disease, taking into account potential pharmacokinetic alterations and drug-induced hepatotoxicity during hepatic metabolism [12]. Furthermore, the clinical management of these patients should be strengthened through a multidisciplinary collaborative model and implemented with more intensive monitoring [201].

## 6. Summary and Limitations

SARS-CoV-2 causes multiple organ damage by binding to the host ACE2 receptor through spike proteins, with liver involvement closely related to high expression of ACE2 in bile duct epithelial cells. The mechanism of liver injury is complex, involving both direct viral effects and multiple indirect effects. Direct infection of bile duct cells by viruses can cause local inflammation and bile stasis; The indirect mechanisms include cytokine storm in immune response, ischemia and hypoxia, oxidative stress NETosis, thrombosis and drug-induced hepatotoxicity. About 15–65% of COVID-19 patients experience liver dysfunction [11]. Mild symptoms include asymptomatic mild to moderate elevation of transaminases, while severe cases may progress to liver failure. Diagnosis requires dynamic monitoring of liver function combined with imaging examinations (excluding other liver diseases). The treatment emphasizes both antiviral and hepatoprotective measures. Treatment with antiviral drugs (such as remdesivir/ritonavir) should take into account liver toxicity and involve promptly adjusting the relevant drugs. In liver protection treatment, supportive therapy is mainly used for mild cases, and glutathione, polyene phosphatidylcholine, etc., can be used in combination for moderate to severe cases. During the treatment process, it is important to pay attention to individualized adjustment plans to avoid metabolic burden.

There are still significant limitations in current research on COVID-19-related liver injury. There is a lack of evidence on whether the virus directly infects liver cells, and the clinical relevance of indirect mechanisms such as cytokine storm and NETosis is not yet clear. The definition and evaluation criteria of liver injury are not unified, and research is mostly based on severe autopsy or retrospective design, neglecting the characteristics of mild cases. There is also some controversy in the field of treatment. There is insufficient evidence for long-term hepatotoxicity and improvement in all-cause mortality of antiviral drugs such as Paxlovid. In terms of prognosis, there is a lack of long-term follow-up data on the liver function outcomes of rehabilitation patients.

Multi-dimensional breakthroughs are needed in the future: Future studies should prioritize establishing a unified definition and grading criteria for liver injury. Mechanism research should clarify evidence of direct virus invasion and validate indirect mechanisms such as NETosis. A multicenter prospective cohort study should be conducted (including patients from different backgrounds), and randomized controlled trials should be designed to evaluate the optimal treatment window and efficacy of antiviral drugs such as Paxlovid. Long-term management should involve the establishment of a follow-up system for recovered patients, including standardized protocols for those with mild COVID-19 that specify follow-up frequency, monitoring indicators, and intervention thresholds. Particular attention should be paid to liver function changes associated with Long COVID, necessitating the development of personalized recovery criteria to support clinical decision-making. Finally, patient education is crucial. Even recovered individuals who had asymptomatic or mild COVID-19—but are experiencing symptoms like fatigue or digestive discomfort—should be considered for liver function screening. These patients should be incorporated into long-term follow-up within both hepatology and integrated post-COVID care systems.

## Figures and Tables

**Figure 1 jcm-14-07228-f001:**
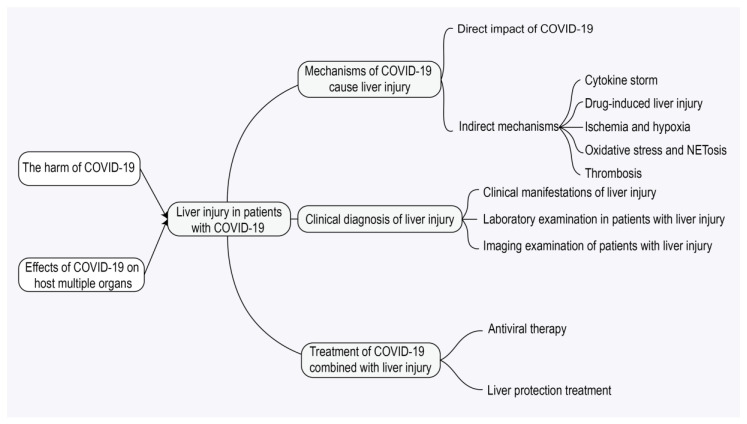
Mind mapping of liver injury associated with coronavirus disease 2019 (COVID-19). This diagram outlines pathogenesis (direct/indirect mechanisms including cytokine storm, drug-induced liver injury [DILI], and neutrophil extracellular trap formation [NETosis]), clinical features, diagnostic approaches (laboratory/imaging), and treatment strategies (antiviral and supportive care) for liver injury in COVID-19 patients. Note: COVID-19, coronavirus disease 2019; DILI, drug-induced liver injury; NETosis, neutrophil extracellular trap formation.

**Figure 2 jcm-14-07228-f002:**
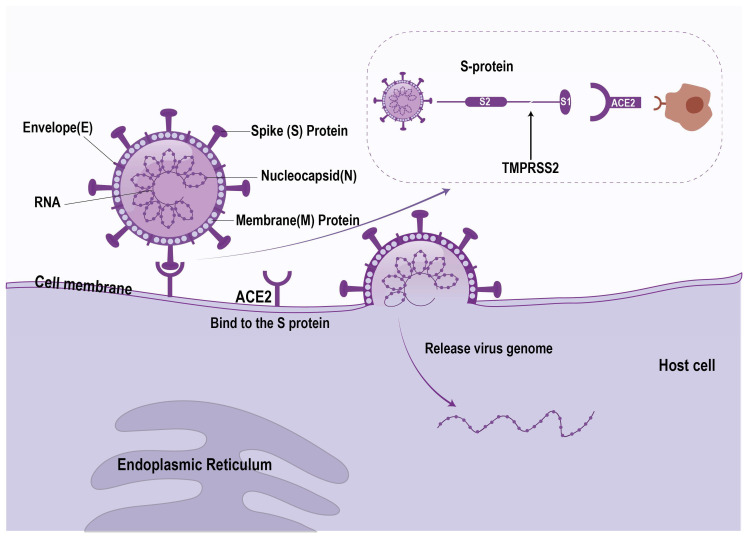
Mechanism of host cell invasion by severe acute respiratory syndrome coronavirus 2 (SARS-CoV-2). The viral entry process begins with the binding of the viral spike (S) protein (comprising S1 and S2 subunits) to the angiotensin-converting enzyme 2 (ACE2) receptor on the host cell surface. This interaction is facilitated by host transmembrane protease serine 2 (TMPRSS2), which cleaves and activates the S protein. Following binding and activation, the viral envelope fuses with the host cell membrane, enabling release of the viral ribonucleic acid (RNA) genome into the cytoplasm. The schematic highlights key viral structural components: membrane (M), envelope (E), nucleocapsid (N), and spike (S) proteins. Note: SARS-CoV-2, severe acute respiratory syndrome coronavirus 2; ACE2, angiotensin-converting enzyme 2; TMPRSS2, transmembrane protease serine 2; RNA, ribonucleic acid; E protein, envelope protein; M protein, membrane protein; N protein, nucleocapsid protein; S, spike protein.

**Figure 3 jcm-14-07228-f003:**
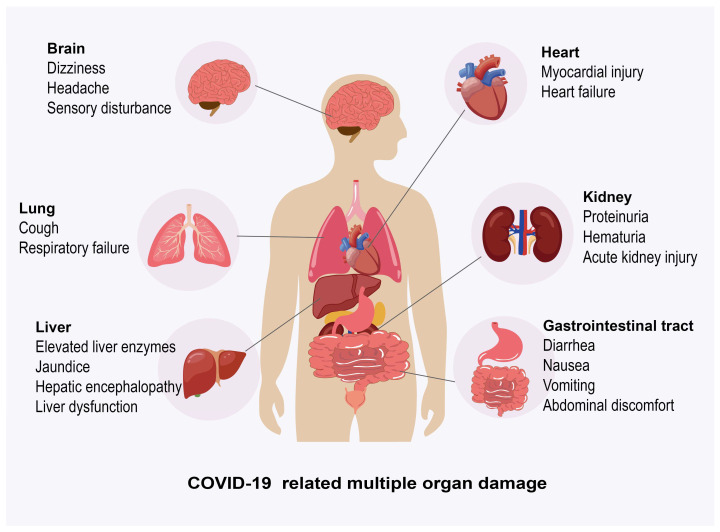
Effects of coronavirus disease 2019 (COVID-19) on multiple host organs. This schematic diagram illustrates the widespread effects of severe acute respiratory syndrome coronavirus 2 (SARS-CoV-2) infection on various organ systems throughout the body. The affected organs and their associated clinical manifestations include lungs (cough, respiratory failure); liver (elevated liver enzymes, jaundice, hepatic encephalopathy, liver dysfunction); gastrointestinal tract (diarrhea, nausea, vomiting, abdominal discomfort); kidneys (proteinuria, hematuria, acute kidney injury); heart (myocardial injury, heart failure); and brain (dizziness, headache, sensory disturbance). Note: COVID-19, coronavirus disease 2019; SARS-CoV-2, severe acute respiratory syndrome coronavirus 2.

**Figure 4 jcm-14-07228-f004:**
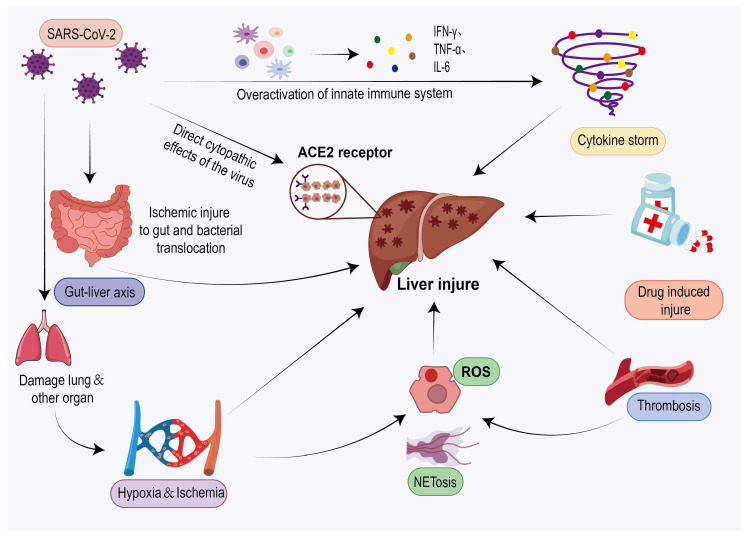
Mechanism of liver injure in coronavirus disease 2019 (COVID-19). Multiple pathways contribute to hepatic injury following severe acute respiratory syndrome coronavirus 2 (SARS-CoV-2) infection. These include direct viral cytopathic effects via angiotensin-converting enzyme 2 (ACE2) receptor binding, cytokine storm (elevated interferon-gamma [IFN-γ], tumor necrosis factor-alpha [TNF-α], and interleukin-6 [IL-6]), drug-induced liver injury(DILI), ischemic/hypoxic damage, reactive oxygen species (ROS) production, neutrophil extracellular trap formation (NETosis), and thrombosis. Gut–liver axis disruption can also lead to hepatic injury. Note: SARS-CoV-2, severe acute respiratory syndrome coronavirus 2; COVID-19, coronavirus disease 2019; ACE2, angiotensin-converting enzyme 2; IFN-γ, interferon-gamma; TNF-α, tumor necrosis factor-alpha; IL-6, interleukin-6; DILI, drug-induced liver injury; ROS, reactive oxygen species; NETosis, neutrophil extracellular trap formation.

**Table 1 jcm-14-07228-t001:** Comparison of the characteristics of each mechanism.

Mechanism	Description of the Specific Mechanism	Target Cells	Core Molecules	Evidence	References
Direct viral infection	The virus infects hepatocytes or cholangiocytes via the ACE2 receptor (cholangiocytes have higher ACE2 expression)	Cholangiocytes, hepatocytes	ACE2 receptor, spike protein, TMPRSS2	Autopsy revealed the presence of viral RNA and proteins in the liver	[18,70,71,72,73,74]
Immune follow-up response (cytokine storm)	Excessive immune response releases pro-inflammatory factors such as IL-6 and TNF-α, which triggers hepatocellular inflammation and necrosis	Macrophages, T cells	IL-6, TNF-α, IFN-γ	Patients have markedly elevated cytokine levels	[12,75,76,77,78]
Hypoxia/ischemic injury	Systemic hypoxemia and hepatic microcirculatory disorders due to COVID-19 cause hepatocyte-ischemic-hypoxic injury	Hepatocytes, hepatic sinusoidal endothelial cells	ROS	Hepatic pathology shows hypoxia/hypoperfusion lesions	[79,80,81]
Oxidative stress and NETosis	SARS-CoV-2 forms a vicious cycle of oxidative damage and NETs release through ROS burst and neutrophil activation	Hepatocytes, neutrophils	ROS, NETs	Mitochondrial dysfunction is found to coexist with microthrombosis/NETs in liver tissue	[82,83,84,85]
thrombosis	The virus induces systemic hypercoagulability and hepatic microvascular thrombosis	Hepatic sinusoidal endothelial cells, platelets	D-dimer, coagulation factor V, fibrinogen	Autopsy reports microthrombosis of the hepatic sinusoids	[81,86,87,88,89,90,91]
Gut–liver axis disorders	Dysbiosis of the intestinal flora leads to endotoxin (LPS) entering the liver, activating Kupffer cells and triggering inflammation	Kupffer cells, intestinal barrier cells	LPS, TLR4/NF-κB pathway	Increased intestinal permeability in patients is associated with liver damage	[92,93,94]
Drug-induced liver injury	Hepatotoxicity of antiviral drugs (e.g., remdesivir), immunomodulators	Hepatocytes	CYP3A4	The incidence of elevated liver enzymes after treatment was 25.4% (57.8% in the lopinavir/ritonavir group)	[95,96,97,98,99,100,101,102,103]

Note: ACE2, angiotensin-converting enzyme 2; COVID-19, coronavirus disease 2019; CYP3A4, cytochrome P450 3A4; DILI, drug-induced liver injury; IFN-γ, interferon-gamma; IL-6, interleukin-6; LPS, lipopolysaccharide; NETs, neutrophil extracellular traps; NF-κB, nuclear factor kappa B; ROS, reactive oxygen species; SARS-CoV-2, severe acute respiratory syndrome coronavirus 2; TLR4, Toll-like receptor 4; TNF-α, tumor necrosis factor-alpha; TMPRSS2, transmembrane protease serine 2.

**Table 2 jcm-14-07228-t002:** Summary Table of RUCAM-Assessed COVID-19-Related Drug-Induced Liver Injury (DILI) Cases.

First Author	Drug(s) Involved	Patient Profile and Liver Injury Manifestations	RUCAM Score and Causality Assessment	Key Conclusions	References
Muhović, D. et al.	Tocilizumab	52-year-old male with severe COVID-19 pneumonia. Developed acute hepatocellular injury 1 day after Tocilizumab administration: ALT 1541 IU/L, AST 1076 IU/L (~40× ULN), no bilirubin elevation.	Score: 8 Causality: Probable	First reported case of RUCAM-verified DILI caused by Tocilizumab in a COVID-19 patient. Other causes (viral, ischemic, etc.) were excluded. Liver enzymes normalized 10 days after drug withdrawal.	[101]
Chen, F. et al.	Multiple drugs	Retrospective analysis of 830 COVID-19 patients found 27.3% (227/830) had abnormal liver biochemistry.	32.6% of patients with abnormal liver tests had a RUCAM score > 3	Suggests a definitive contribution of drug factors to COVID-19-associated liver injury. RUCAM helps avoid misdiagnosis of non-drug-related liver injury.	[141]
Delgado, A. et al.	Multiple drugs	DILI was detected in 160 hospitalized COVID-19 patients. Most cases were hepatocellular (57.5%) and mild (87.5%), often with polypharmacy (average 14.7 drugs per patient).	51.2%: Probable (Score ≥ 6) 48.8%: Possible (Score 3–5)	Remdesivir had the highest incidence rate of DILI (992.7 per 10,000 DDD). Combining RUCAM with LTT testing may improve diagnostic accuracy for DILI involving immune mechanisms.	[138]
Jothimani, D. et al.	Dabigatran	51-year-old male developed severe jaundice (Total Bilirubin 39.1 mg/dL) and pruritus 3 weeks after starting Dabigatran post-COVID-19 discharge. Liver biopsy suggested cholestatic DILI.	Score: 7 Causality: Probable	Other causes (viral, autoimmune, biliary obstruction) were excluded. Suggests Dabigatran may cause cholestatic DILI, potentially via an idiosyncratic reaction.	[142]
Kumar, P. et al.	Favipiravir	3 COVID-19 patients developed liver injury post-treatment (1 hepatocellular, 2 cholestatic), with significant elevations in ALT/AST/ALP.	Score: 7 Causality: Probable	First case series report of Favipiravir-induced DILI. Liver function recovered after drug cessation, indicating its hepatotoxicity may be reversible.	[100]
Yamazaki, S. et al.	Favipiravir	A 73-year-old male with severe COVID-19 developed cholestatic liver injury (elevated ALP, γ-GTP, TBIL) after Favipiravir treatment.	Score: 6 Causality: Probable	First reported case of Favipiravir-induced cholestatic liver injury. Underlying liver disease and high dosage were suggested as potential risk factors.	[143]

Note: This table synthesizes and summarizes the key findings from the six primary studies of RUCAM-assessed DILI in COVID-19 patients that were systematically reviewed by Teschke et al. [16]. The data are presented here with a focus on the implicated drugs and their causality assessment. The original studies are referenced in the rightmost column. Abbreviations: ALT, alanine aminotransferase; AST, aspartate aminotransferase; ALP, alkaline phosphatase; TBIL, total bilirubin; γ-GTP, gamma-glutamyl transferase; ULN, upper limit of normal; LTT, lymphocyte transformation test.

**Table 3 jcm-14-07228-t003:** Summary of Incidence Rates and Definition Variability of Liver Injury in COVID-19 Patients Across Different Studies.

Study Author	Study Type	Sample Size	Incidence of Liver Injury (Parameter Type)	Key Definition of Liver Injury (Core Differences)	References
Hundt et al.	Single-center Retrospective Cohort Study	*n* = 1827	At admission: AST elevation: 66.9% ALT elevation: 41.6% ALP elevation: 13.5% TBIL elevation: 4.3%	Based on hospital laboratory reference ranges: AST > 33 U/L, ALT > 34 U/L, ALP > 122 U/L, TBIL > 20.5 μmol/L	[155]
McConnell et al.	Multicenter Retrospective Study + Experimental Validation	*n* = 3780	ALT ≥ 3 × ULN: 12.7% (481/3780)	ALT ≥ 3 × ULN (Based on local laboratory standards)	[86]
Phipps et al.	Multicenter Retrospective Cohort Study	*n* = 2273	ALT Grading: Mild (>ULN but <2×): 45% Moderate (2–5×): 21% Severe (>5×): 6.4% (95% CI: Not provided)	Grading based on ALT multiples (ULN = 50 U/L)	[44]
Chen et al.	Single-center Retrospective Study	*n* = 99	ALT or AST elevation: 43.4% (43/99) (95% CI: Not provided)	ALT or AST > Upper Limit of Normal (ALT > 50 U/L, AST > 40 U/L)	[156]
Lei et al.	Multicenter Retrospective Study	*n* = 5771	ALT elevation: Common in severe cases, median 26.0 U/L (IQR 17.0–45.0) AST elevation: Median in severe cases 31.0 U/L (IQR 21.0–46.0) ALP: Mostly within normal range TBIL: Mild elevation, median 10.6 μmol/L (IQR 7.9–15.0)	Acute liver injury defined as: ALT > 3 × ULN Liver enzyme elevation defined as: >ULN	[157]
Mao et al.	Systematic Review (SR)	35 studies, *n* = 6686	Pooled prevalence of liver dysfunction: 19% (95% CI: 9–32; range 1–53; I^2^ = 96%). Compared with non-severe patients, severe COVID-19 patients had a higher incidence of liver dysfunction, including ALT elevation (OR: 1.89, 95% CI: 1.30–2.76; *p* = 0.0009; I^2^ = 10%) and AST elevation (OR: 3.08, 95% CI: 2.14–4.42; *p* < 0.00001; I^2^ = 0%).	Based on the indicator abnormality criteria set by the respective included studies.	[158]
Wijarnpreecha et al.	Systematic Review (SR)	64 studies, *n* = 11,245	Approximately 25% of COVID-19 patients had elevated liver enzymes. The prevalence of specific parameter elevations were AST 23.2%, ALT 21.2%, Total Bilirubin 9.7%, GGT 15.0%, ALP 4.0%. The prevalence of AST elevation was higher in severe cases (45.5%) compared to non-severe cases (15.0%). Up to 37.6% of COVID-19 patients had concurrent CLD.	Based on the indicator abnormality criteria set by the respective included studies.	[159]
Wang et al.	Single-center Retrospective Study	*n* = 105	56.2% of patients had abnormal ALT, AST, or TBIl during the disease course (91.4% of abnormal parameters were ≤3 × ULN); the rate of concurrent ALT + AST elevation was higher in the severe group (46.2%) than in the mild group (12.7%), and 34.6% of severe patients developed new ALT abnormalities after admission.	Defined as exceeding the normal range for ALT (M: 9.0–50.0 U/L; F: 7.0–40.0 U/L), AST (M: 15.0–40.0 U/L; F: 13.0–35.0 U/L), TBil (0–18.8 μmol/L), graded by 1 × ULN, 2 × ULN, 3 × ULN.	[160]

Note: ALP, alkaline phosphatase; ALT, alanine aminotransferase; AST, aspartate aminotransferase; CI, confidence interval; CLD, chronic liver disease; COVID-19, coronavirus disease 2019; F, female; GGT, gamma-glutamyl transferase; IQR, interquartile range; M, male; OR, odds ratio; SR, systematic review; TBIL, total bilirubin; ULN, upper limit of normal.

## Data Availability

Not applicable.

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
