# Peer review of "COVID-19 Infection, Drugs, and Liver Injury"

_jcm, 2025, doi:10.3390/jcm14207228_

Round 1
Reviewer 1 Report
Comments and Suggestions for Authors
The manuscript is presented as a review, but it does not detail the literature search strategy (databases, keywords, timeframes, inclusion/exclusion criteria). This limits reproducibility and rigor.
I recommend adding a methodology section that explains how studies were selected, even if a strict systematic review was not performed.
The article devotes extensive space to pathophysiological mechanisms (direct infection, NETosis, oxidative stress, etc.), which are very interesting, but their relevance to current clinical practice is not well prioritized.
I suggest clearly distinguishing which of these mechanisms are supported by solid clinical evidence and which remain within the realm of experimental hypotheses.
Different incidence rates are mentioned (15–65%) depending on the studies, but the differences in the definition of liver injury across series are not sufficiently discussed.
It would be useful to include a table or summary of the diagnostic criteria employed in the studies and to propose the need for standardization.
The section on antivirals and hepatoprotective agents is interesting, but in many cases it lacks a critical appraisal of the evidence. For example, remdesivir and Paxlovid are discussed in general terms, without addressing the magnitude of benefit or comparative hepatic risks.
I recommend organizing the section according to a framework of “drugs with proven efficacy / probable efficacy / inconclusive evidence” and including levels of evidence.
The article devotes little space to the implications for patients with cirrhosis, liver transplantation, or chronic liver diseases (NAFLD, prior viral hepatitis). Given the clinical impact, it would be essential to expand this aspect.
Reviewer 2 Report
Comments and Suggestions for Authors
The article titled COVID-19 Infection and Liver Injury presents a multifactorial analysis of the potential impact of SARS-CoV-2 infection on the development of liver injury.
The manuscript thoroughly discusses the possible pathophysiological mechanisms associated with coronavirus infection and COVID-19 in relation to the liver. It addresses the potential direct effects of the virus on hepatocytes, the role of the cytokine storm, oxidative stress, ischemia and hypoxia, vascular complications, disturbances of the microbiota, as well as the possible hepatotoxic effects of drugs used in COVID-19 therapy.
Diagnostic methods, both clinical, laboratory, and imaging, are also described.
The article is well written, and the references are up to date.
The topic is highly relevant, especially in the context of the currently observed post-pandemic increase in the number of patients with chronic liver diseases, as well as the ongoing circulation of SARS-CoV-2 in the population, its new subvariants, and the associated periodic infections—both asymptomatic and symptomatic.
It is worth adding that even asymptomatic or mildly symptomatic SARS-CoV-2 infection may be associated with organ complications, including hepatic injury, as well as post-COVID syndrome (long COVID). Liver-related complications may also appear many months, or even years, after SARS-CoV-2 infection through immunological mechanisms.
Additionally, in the section on diagnostic methods, it would be advisable to mention the role of liver elastography using FibroScan in assessing liver injury—fibrosis and steatosis—as well as liver biopsy.
The article also suggests that patients after COVID-19 should be included in hepatological care and screening within post-COVID management, including long-term follow-up.
Reviewer 3 Report
Comments and Suggestions for Authors
Interesting approach on an important clinical issue but some shortcomings prevail.
Major points:
- Many causes for increased liver tests (LT) are mentioned by you that are speculative and not based on evidence because causality assessment methods were not applied or not applicable. So, be more critical now in the revised version. Among these causes lacking evidence are: viral infection (via ACE2 receptors), immune follow-up response (cytokine storm), ischemic/hypoxic liver injury, thrombosis, oxidative stress, and NETosis, and enterohepatic axis.
- You correctly mentioned DILI as possible cause but did not provide evidence thereof, another major shortcoming of your paper. Important are cases of DILI, assessed for causasilty either by the the original RUCAM published 1993, or now better evaluated by the updated RUCAM of 2016. RUCAM should also be included under keywords.
- You must re-write the DILI section. New discussion and quotation should focus on DILI cases with verified causality by RUCAM, see for instance; doi: 10.3390/ijms23094828.
- Besides, you should mention in the introduction that, although many causes of increased LTs remain unclear, clarity exists for DILI as cause because RUCCAM excluded a priori alternative causes and strictly considers drugs as causes. Of course, as many patients received drugs, it it important to differentiate non-drug causes from drug causes,
- For reasons of completeness, add a list of RUCAM based DILI cases: Muhović, D.; Bojović, J.; Bulatović, A.; Vukčević, B.; Ratković, M.; Lazović, R.; Smolović, B. First case of drug-induced liver injury associated with the use of tocilizumab in a patient with COVID-19. Liver Int. 2020, 40, 1901–1905 Chen, F.; Chen, W.; Chen, J.; Xu, D.; Xie, W.; Wang, X.; Xie, Y. Clinical features and risk factors of COVID-19-associated liver injury and function: A retrospective analysis of 830 cases. Hepatol.2020, 21, 100267. [Google Scholar] [CrossRef] [PubMed] Delgado, A.; Stewart, S.; Urroz, M.; Rodríguez, A.; Borobia, A.M.; Akatbach-Bousaid, I.; González-Muñoz, M.; Ramírez, E. Characterisation of Drug-Induced Liver Injury in Patients with COVID-19 Detected by a Proactive Pharmacovigilance Program from Laboratory Signals. Clin. Med.2021, 10, 4432. [Google Scholar] [CrossRef] [PubMed]
Jothimani, D.; Vij, M.; Sanglodkar, U.; Patil, V.; Sachan, D.; Narasimhan, G.; Kaliamoorthy, I.; Rela, M. Severe Jaundice in a COVID-19 Patient–Virus or Drug? Clin. Exp. Hepatol.2021, 11, 407–408. [Google Scholar] [CrossRef]
Kumar, P.; Kulkarni, A.; Sharma, M.; Raon, P.N.; Reddy, D.N. Favipiravir-induced liver injury in patients with coronavirus disease 2019. Clin. Transl. Hepatol.2021, 9, 276–278. [Google Scholar] [CrossRef]
Yamazaki, S.; Suzuki, T.; Sayama, M.; Nakada, T.-A.; Igari, H.; Ishii, I. Suspected cholestatic liver injury induced by favipiravir in a patient with COVID-19. Infect. Chemother.2020, 27, 390–392. [Google Scholar] [CrossRef]
6. Title is too vague. Consider to make your title more promotial: COVID-19 Infection, drugs, and liver injury.
Or even better and more specifically: COVID-19 infection and role of drugs in liver injury as evlated for causality using RUCAM.
Actually, only drugs are evidenced causes and nothing else.
7. Minor: Expand in abstract: NETosis. At one place you mentioned the filling term etc, please add instead more details.
Round 2
Reviewer 3 Report
Comments and Suggestions for Authors
Thank you for attempts to revise. Although some progress is evident, major methodology shortcomings persist but can easily be corrected.
Major points:
- In your covering letter under the third point, you mentioned that "we have also cited the key study you recommended (doi: 10.3390/ijms23094828) to strengthen the evidence. However, despite this note, the report in question was not mentioned and not discussed in the text, nor was it mentioned in the reference list. This must and can easily be done in the re-revision. 2. Introduction, new inserts on RUCAM. While the inserts are good, correct references are missing for RUCAM, especially for the original RUCAM of 1993 but also of the updated RUCAM of 2016 which is mentioned as ref# 136 later on and must be moved to the introduction. In this context, your ref#14 must be replaced by the key study mentioned above under 1. 3. Table 2 is not informative unless supplied by RUCAM scores, can easily be done, otherwise you better remove the table. Is impact on the right term. 4. Figures and tables, ensure that in the legends all abbreviations are expanded. 5. Include RUCAM under the Abbreviation section. 6. Your Table 3 is seemingly a plagiarism retrieved from the key study as mentioned under point 1 above. You must give appropriate credit. 7. Regarding section of liver biopsy, this invasive diagnostic approach is only indicated if the patient has a benefit. It is certainly not advised to stage the disorder. 8. Treatment of liver injury is speculative regarding hepatoprotectives. Efficacy was never based on evidence, so you better remove or provide data of RCTs.
None.
Round 3
Reviewer 3 Report
Comments and Suggestions for Authors
Thank you for perfect evised paper.